# EDTA Chelation Therapy in the Treatment of Neurodegenerative Diseases: An Update

**DOI:** 10.3390/biomedicines8080269

**Published:** 2020-08-03

**Authors:** Alessandro Fulgenzi, Daniele Vietti, Maria Elena Ferrero

**Affiliations:** Department of Biomedical Sciences for Health, Università degli Studi di Milano, Via Mangiagalli, 31, 20133 Milan, Italy; alessandro.fulgenzi@unimi.it (A.F.); danielevietti@gmail.com (D.V.)

**Keywords:** EDTA chelation therapy, neurodegenerative diseases, metal detoxification

## Abstract

We have previously described the role played by toxic-metal burdens in the etiology of neurodegenerative diseases (ND). We herein report an updated evaluation of toxic-metal burdens in human subjects affected or not affected by ND or other chronic diseases. Each subject underwent a chelation test with the chelating agent calcium disodium ethylenediaminetetraacetic acid (CaNA_2_EDTA or EDTA) to identify the presence of 20 toxic metals in urine samples using inductively coupled plasma mass spectrometry. Our results show the constant presence of toxic metals, such as lead, cadmium, cesium, and aluminum, in all examined subjects but the absence of beryllium and tellurium. Gadolinium was detected in patients undergoing diagnostic magnetic resonance imaging. The presence of toxic metals was always significantly more elevated in ND patients than in healthy controls. Treatment with EDTA chelation therapy removes toxic-metal burdens and improves patient symptoms.

## 1. Introduction

Multiple mechanisms are involved in the pathogenesis of neurodegenerative diseases (ND), and knowing how these mechanisms work is of paramount relevance to identify a proper therapeutic strategy. The four most important ND, where both genetic predisposition and environmental factors play important roles, are Parkinson’s disease (PD), Alzheimer’s disease (AD), multiple sclerosis (MS), and amyotrophic lateral sclerosis (ALS). We have previously described the possible major causes and the related mechanisms involved in direct (toxic metals, air pollution, air and electromagnetic fields, pesticides, neurotoxins, and pathogens) and indirect (proinflammatory cytokines and free oxygen radical productions) neurotoxicity associated to ND [1]. Accumulated evidence of toxic metal cellular damage is now disposable. In particular, recent advances in understanding the role of mitochondrial dysfunction in the pathophysiology of both sporadic and familial PD have already been discussed [2]. More generally, mitochondrial dysfunction is present in ND due to an excessive production of reactive nitric oxide-dependent species, which can trigger post-translational protein modification [3]. Bioenergetic deficits related to mitochondrial dysfunction might be responsible for neuron death and the clinical expression of dementia and are possibly associated to late-onset AD [4]. In addition, the function of some neurotrophic receptors, and their involvement in the pathogenesis, diagnosis, and therapy of PD and AD, has been shown [5].

Moreover, the role of intestinal microbiota in ND has been discussed on more than one occasion [6]. Changes in intestinal microbiota, with consequent microorganism-induced modifications in both intestinal and blood-brain barrier permeability, have been linked to an increased risk of developing AD, PD, and ALS. Furthermore, there is evidence showing a correlation between obesity and the development of AD and PD [7]. Indeed, obese patients frequently display type 2 diabetes mellitus (characterized by neuropathies); insulin resistance is related to dementia, while proinflammatory cytokines in adipose tissue contribute to neuroinflammation [7]. The role played by the inflammatory process in the pathogenesis of neurodegeneration, particularly in the elderly, can be explained by the link between inflammation and mental-function impairment [8]. Alongside these important causes of neuron damage or death, we have focused our studies on the role of toxic metals in the pathogenesis of ND and have described the molecular mechanisms of each toxic metal leading to impaired biological functions in multiple organs, which are cumulatively related to the excessive production of detrimental free oxygen/nitrogen radicals [9].

Many epidemiological studies suggest a role of chronic exposure to toxic metals in the development and propagation of cardiovascular disease [10] and in the generation of vascular complications, especially in diabetic patients [11]. Moreover, it has been shown an association between PD and an exposure to metals such as mercury, lead, manganese, copper, iron, aluminum, bismuth, thallium, and zinc [12] and the potential role of mercury in AD [13]. Exposure to heavy metals such as cadmium and arsenic correlates with glutathione-S-transferase polymorphism in Iranian MS patients, due to the enzyme’s ability to remove toxic products [14]. Overall, these findings provide the rationale for the management of ethylenediaminetetraacetic acid (EDTA) chelation therapy [9] as a successful option in the treatment of ND and other diseases associated with metal burdens [1,9]. Notably, chelation therapy has been recently demonstrated a well-tolerated and effective treatment method for post-myocardial infarction patients [15]. The present report is an update and extension on the relationship between toxic-metal burdens and ND, non-ND, and healthy controls [16], with particular focus on the profile of toxic metals in ALS patients. Our study, aimed to investigate the potential and the efficacy of chelation therapy in the cure of subjects affected by toxic metal burden, encouraging its employment in removing toxic-metal poisoning and related symptoms, also through an extensive clinical description of a representative case.

## 2. Materials and Methods

### 2.1. Patients

We studied 379 doctors’ office patients, age ranging from 13 to 87 years. They gave their consent to undergo chelation therapy, with the chelating agent calcium disodium ethylenediaminetetraacetic acid (EDTA) as unique therapy to treat the disease. Many patients were affected by ND, while others were affected by non-ND (cardiovascular disease, fibromyalgia, rheumatoid arthritis, and peripheral neuropathies); other subjects exposed to occupational or environmental toxic metals, but unaffected by ND or non-ND, acted as healthy controls. The experimental protocol was approved by Milan University’s Ethics Advisory Committee (number 64/14). All procedures were performed in accordance with the ethical standard of the responsible committee of human experimentation and with the Helsinki declaration revised in 2000. Informed consent was obtained from each patient included in the study.

### 2.2. Study Design

All of the patients carried out a “chelation test” (see below) to investigate their possible toxic-metal burdens. Thereafter, they underwent chelation therapy for almost three months. The chelation test was repeated after ten applications to assess the body-burden modifications. The patients were monitored throughout therapy. 

### 2.3. Chelation Test

This was performed as previously described [16]. Briefly, EDTA (2 g) diluted in 500-mL physiological saline (Farmax srl, Brescia, Italy) was slowly (over 2 h) administered intravenously to patients. They were invited to collect urine samples before and for 12 h after the initial intravenous EDTA treatment. Urine samples accurately enveloped in sterile vials were sent to the Laboratory of Toxicology (Doctor’s Data Inc., St. Charles, IL, USA) for analysis, as previously reported [16]. Samples were acid-digested with certified metal-free acid, diluted with ultrapure water, and examined via inductively coupled plasma mass spectrometry (ICP-MS), a reliable method to reduce interference. Urine standards, both certified and in-house, were used for quality control and data validation. To avoid a potential error due to fluid intake and sample volume, the results were reported in micrograms (µg) per g of creatinine. Patients showing toxic-metal burdens at the chelation test underwent chelation therapy.

### 2.4. Chelation Therapy

Chelation therapy was performed by a weekly intravenous infusion of 2-g EDTA in physiological saline. Each patient underwent almost 30 chelation therapy applications. After ten applications, a further chelation test was carried out. Toxic-metal burden values in urine samples are referred to as mineralograms.

### 2.5. Toxic-Metal Analysis

Twenty toxic metals were analyzed: aluminum (Al), antimony (Sb), arsenicum (As), barium (Ba), beryllium (Be), bismuth (Bi), cadmium (Cd), cesium (Cs), gadolinium (Gd), lead (Pb), mercury (Hg), nickel (Ni), palladium (Pd), platinum (Pt), tellurium (Te), thallium (Tl), thorium (Th), tin (Sn), titanium (Ti), tungsten (W) and uranium (U). Gadolinium is frequently used as a contrast agent in magnetic resonance imaging to diagnose ND.

### 2.6. Statistical Analysis

Results were expressed as standard error mean of mean (mean ± SEM). They were analyzed using *t*-tests. Statistical tests were two-sided, and significance was assumed at *p* < 0.05. We used IBM SPSS Statistics. We used also ANOVA to compare the groups (HC vs. ND, non-ND, and ALS).

## 3. Results

### 3.1. Patient Characteristics

The patient population was classified as ND, non-ND, and healthy controls (HC) (see Materials and Methods section) (Figure 1). We examined 179 men (mean age = 50.61 years) and 200 women (mean age = 50.82 years). The majority of ND patients were affected by MS.

### 3.2. Percentage of Patients Affected by Each Toxic-Metal Burden vs. Total Poisoned Population

Figure 2 shows the percentage of ND and HC subjects within the total population: each of the twenty metals known as toxic are analyzed: aluminum (Al), antimony (Sb), arsenicum (As), barium (Ba), beryllium (Be), bismuth (Bi), cadmium (Cd), cesium (Cs), gadolinium (Gd), lead (Pb), mercury (Hg), nickel (Ni), palladium (Pd), platinum (Pt), tellurium (Te), thallium (Tl), thorium (Th), tin (Sn), tungsten (W), and uranium (U). With the exception of thorium, all ND patients presented a more elevated toxic-metal burden compared with HC. Of note, only one HC patient was intoxicated by thorium, possibly due to accidental exposure.

### 3.3. Toxic-Metal Burdens in Patient Urine Samples Following Chelation Test

Patient urine samples collected before the chelation test did not reveal significant toxic-metal contents (data not shown). Toxic-metal burden values assessed in the urine samples taken from patients following the chelation test *(*e.g., after the first treatment with EDTA*)* are shown in Table 1. The cut-off represents the limit values of toxic metals, as higher values indicate toxicity. Patients with toxic-metal values above the cut-off are reported in column 3 of Table 1 and indicated as A for each toxic metal in the total population (TP = all patients examined). The percentage of intoxication by each toxic metal with respect to the TP is reported in column 4. Columns 5 and 6 respectively show the mean and standard error of the mean (SEM) of the metal level values above the cut-off. Columns 7 and 8 show the number of ND patients burdened with each toxic metal and the percentage of those patients vs. A. Columns 9 and 10 show the mean values of toxic-metal levels above the cut-off and the SEM in ND patients. Albeit MS is the most frequent ND, we here consider the ALS patients separately from the others ND patients to assess whether they exhibit a different profile of toxic metals in both quality and quantity. Columns 11 and 12 show the number of ALS patients affected by toxic-metal burdens (i.e., levels of toxic metals above the cut-off) and their percentage vs. A. Columns 13 and 14 show the mean values and SEM of toxic-metal levels above the cut-off in ALS patients. Columns 14 and 15 show the number of patients affected by each toxic-metal burden and their percentage vs. A in non-ND patients. Columns 16 and 17 show the mean values of toxic-metal levels above the cut-off and SEM in non-ND patients. Columns 18 and 19 show the number of patients affected by toxic-metal burdens relative to each toxic metal and their percentage vs. A in HC patients. Columns 20 and 21 show the mean values and SEM of toxic-metal levels above the cut-off in HC patients.

Aluminum. Patients affected by Al burdens totaled 135, representing the 36.4% of the TP. The cut-off value for Al is 25 µg/g creatinine, measured in the urine samples, and the mean value of Al > cut-off was 41.95 ± 1.74. Neurodegenerative disease patients affected by Al burdens totaled 47 (35% of A), with a mean value of Al > cut-off = 46.70 ± 2.90. Of some ND patients affected by ALS presenting Al burdens: they were 14 (10% of A), with a mean value of Al > cut-off = 56.86% ± 2.21. Non-ND patients affected by Al burdens totaled 65 (48% of A), with a mean value of Al > cut-off = 47.67 ± 2.11. Patients classified as HC affected by Al burdens totaled 14 (10% of A), with a mean value of Al > cut-off = 32.43% ± 0.53.

Antimony. Five patients were affected by Sb burdens, representing 1.6% of the TP. The cut-off value for Sb was 0.30 µg/g creatinine in the urine samples, and the mean value of Sb > cut-off was 1.25 ± 0.27. Neurodegenerative patients affected by Sb burdens totaled four (80% of A), with a mean value of Sb > cut-off = 1.53 ± 0.73. One patient affected by ALS presented a Sb burden (20% of A), with a value of Sb > cut-off = 0.07. Only one non-ND patient was affected by a Sb burden (20% vs. A), with the value of Sb > cut-off = 1.80. No HC patient was affected by a Sb burden.

Arsenic. Patients affected by As burdens totaled 55 (15.3% of TP). The cut-off value for As was 108.00 µg/g creatinine in the urine samples, with a mean value of As > cut-off = 252.07 ± 13.85. Nineteen ND patients were affected by As burdens (35% of A), with a mean value of As > cut-off = 269.47 ± 18.34. One ALS patient only was affected by an As burden, with a value > cut-off = 77.95%. Twenty-five non-ND patients were affected by As burdens (45% vs. A), with a mean value of As > cut-off = 317.17 ± 21.35. Patients classified as HC affected by As burdens were nine (15% vs. A), with a mean value of As > cut-off = 288.89 ± 24.73.

Barium. Thirteen patients were affected by Ba burdens (3.7 of TP). The cut-off value for Ba was 7.00 µg/g creatinine, and the mean value of Ba > cut-off was = 76.43 ± 7.42. Four ND patients were affected by Ba burdens (31% of A), with a mean value of Ba > cut-off of 52.83 ± 17.49. Two ALS patients were affected by Ba burdens (25% of A), with a not statistically significant mean value of Ba > cut-off. Six non-ND patients and two HC patients only were affected by Ba burdens.

Beryllium. No patients displayed Be intoxication.

Bismuth. One patient only was affected by a Bi burden and was an ALS patient.

Cadmium. Patients affected by Cd burdens totaled 346 (92.9% of TP). The cut-off value for Cd was 0.80 µg/g creatinine, and the mean value of Cd > cut-off was = 3.49 ± 0.16. Neurodegenerative patients affected by Cd burdens totaled 107 (31% of A), with a mean value of Cd > cut-off = 3.96 ± 0.21. Patients affected by ALS and bearing Cd burdens were 24 (7% of A), with a mean value of Cd > cut-off = 3.08 ± 0.16. Non-ND patients affected by Cd burdens totaled 179 (52% of A), with a mean value of Cd > cut-off = 4.30 ± 0.14). Finally, HC totaled 46 (15% of A), with a mean value of Cd > cut-off = 2.52 ± 0.08).

Cesium. Patients affected by Cs burdens totaled 177 (47.5% of TP). The cut-off value for Cs was 9.00 µg/g creatinine, and the mean value of Cs > cut-off was = 14.63 ± 0.53. Neurodegenerative disease patients affected by Cs burdens were 59 (33% of A), with a mean value of Cs > cut-off = 16.00 ± 0.82. Patients affected by ALS with Cs burdens were 16 (9% of A), and the mean value of Cs > cut-off was = 8.78 ± 0.64. Non-ND patients affected by Cs burdens totaled 86 (49% of A), with a mean value of Cs > cut-off 16.78 ± 0.53. Healthy controls with Cs burdens totaled 22 (12% of A), with a mean value of Cs > cut-off = 12.88 ± 0.25.

Gadolinium. Patients affected by Gd intoxication totaled 172 (45.9% of TP). The cut-off value for Gd was 0.30 µg/g creatinine, and the mean value of Gd > cut-off was 31.19 ± 5.41. Neurodegenerative disease patients affected by Gd burdens were 88 (51% of A), with a mean value of Gd > cut-off = 41.55 ± 6.96. Seven patients affected by Gd intoxications were ALS patients (4% of A), with a mean value of Gd > cut-off = 58.83 ± 0.81. Non-ND patients affected by Gd intoxications were 65 (38% of A), with a mean value of Gd > cut-off = 8.48 ± 4.34. Healthy control patients affected by Gd intoxications were 11 (6% of A), with a mean value of Gd > cut-off = 2.62 ± 0.18.

Lead. Patients affected by Pb intoxications totaled 370 (99.7% of TP). The cut-off value for Pb was 2.0 µg/g creatinine, and the mean value of Pb > cut-off was 26.76 ± 1.56. Neurodegenerative disease patients affected by Pb burdens totaled 115 (31% of A), with a mean value of Pb > cut-off = 28.00 ± 1.91. Patients with ALS affected by Pb burdens were 24 (6% of A), with a mean value of Pb > cut-off = 22.92 ± 1.75. Non-ND patients affected by Pb burdens were 192 (52% of A), with a mean value of Pb > cut-off = 38.79 ± 1.53. Healthy controls affected by Pb burdens totaled 50 (14% of A), with a Pb value > cut-off = 20.16 ± 0.77.

Mercury. Patients affected by Hg intoxications totaled 18 (5% of TP). The cut-off value for Hg was 3.00 µg/g creatinine, and the mean value of Hg > cut-off was 7.58 ± 0.66. Neurodegenerative disease patients affected by Hg burdens totaled six (33% of A), with a mean value of Hg > cut-off = 7.17 ± 1.78. Of the ND patients, three ALS patients (17% of A) were intoxicated by Hg, with a mean value of Hg > cut-off = 1.55 ± 0.12. Non-ND patients bearing Hg burdens were 11 (61% of A), with a mean value of Hg > cut-off = 7.53 ± 2.57. Only one HC patient was affected by a Hg burden.

Nickel. Patients affected by Ni intoxications totaled 58 (16.1% of TP). The cut-off value for Ni was 10.00µg/g creatinine, and the mean value of Ni > cut-off was 16.84 ± 0.99. Patient with ND affected by Ni burdens were 24 (41% of A), with a mean value of Ni > cut-off = 21.50 ± 3.07. Seven patients with ALS were affected by Ni burdens (12% of A), with a mean value of Ni > cut-off = 8.80 ± 0.96. Non-ND patients affected by Ni burdens were 29 (50% of A), with a mean value of Ni > cut-off = 21.93 ± 0.58. Healthy controls bearing Ni burdens were five (9% of A), with a mean value of Ni > cut-off = 12.00 ± 0.16.

Palladium. Patients affected by Pd intoxications were 13 (3.4% of TP). The cut-off value for Pd was 0.30 µg/g creatinine, and the mean value of Pd > cut-off was 0.48 ± 0.04. Eight ND patients were affected by Pd burdens (62% of A), with a mean value of Pd > cut-off = 0.48 ± 0.05. Only one ALS patient was affected by a Pd burden. Five non-ND patients displayed Pd burdens (38% of A), with a mean value of Pd > cut-off = 0.65 ± 0.03. Only one HC patient displayed a Pd burden. 

Platinum. Only two patients were affected by Pt intoxications and were non-ND patients. 

Tellurium. No patient was affected by a Te intoxication.

Thallium. Patients affected by Tl intoxications totaled 55 (14.8% of TP). The cut-off value for Tl was 0.50 µg/g creatinine, with a mean value of Tl > cut-off = 1.20 ± 0.06. Patients affected ND and by Tl burdens were 16 (29% of A), with a mean value of Tl > cut-off = 1.13 ± 0.12. Eight ALS patients (15% of A) were affected by Tl burdens. Thirty-one non-ND patients were affected by Tl burdens (56% of A), with a mean value of Tl > cut-off = 1.15 ± 0.12. Five HC patients were affected by Tl burdens (9% of A), with a mean value of Tl > cut-off = 0.72 ± 0.02.

Thorium. Only one patient was affected by a Th intoxication and was a HC patient.

Tin. Patients affected by Sn burdens were four (11% of TP). The cut-off value for Sn was 9.00 µg/g creatinine, and the mean value of Sn > cut-off was 24.00 ± 1.47. No ND patients nor HC patients were affected by Sn intoxications. Three non-ND patients were affected by Sn intoxications (75% of A), with a mean value of Sn > cut-off = 48.00 ± 2.29. 

Tungsten. Patients affected by W intoxications were 35 (9.5% of TP). The cut-off value for W was 0.40 µg/g creatinine, and the mean value of W > cut-off was 1.18 ± 0.18. Neurodegenerative patients affected by W burdens totaled 14 (40% of A), with a mean value of W > cut-off = 2.06 ± 0.82. Seven ALS patients were affected by W intoxications. Nineteen non-ND patients were affected by W intoxications (54% of A), with a mean value of W > cut-off = 2.88 ± 0.04. Finally, three HC patients were affected by W intoxications (9% of A), with a mean value of W > cut-off = 0.60 ± 0.03. 

Uranium. Patients affected by U intoxications were 23 (6.3% of A). The cut-off for U was 0.03 µg/g creatinine, with a mean value of U > cut-off = 0.39 ± 0.24. Neurodegenerative disease patients affected by U burdens were six (26% of A), with a mean value of U > cut-off = 0.25 ± 0.076. Only one ALS patient was affected by a U burden. Non-ND patients affected by U burdens were 15 (65% of A), with a mean value of U > cut-off = 0.31 ± 0.299. Two HC patients were affected by U burdens (9% of A), with a mean value of U > cut-off = 0.09 ± 0.004. 

The results obtained deserve consideration. Firstly, no one was affected by beryllium or tellurium intoxications. Patients were mainly intoxicated by lead, cadmium, cesium, gadolinium, aluminum, and, to a lesser extent, nickel, arsenic, thallium, and tungsten. Patients affected by ND were intoxicated by lead, cadmium, gadolinium, cesium, and aluminum. In particular, those affected by ALS were intoxicated by lead, cadmium, cesium and aluminum. Gadolinium intoxications were related to the elevated number of MRI examinations undergone by ND patients, especially by MS patients and also by some non-ND patients. Patients affected by non-ND also showed elevated levels of lead, cadmium, cesium, gadolinium, and aluminum. Healthy controls displayed elevated levels of lead, cadmium, and cesium. All examined ALS patients were intoxicated by lead and cadmium and, to a lesser extent, by gadolinium and aluminum. Maximum levels of lead and cadmium intoxications were reached by non-ND patients. The levels of aluminum, cadmium, cesium, lead, and nickel were significantly more elevated in ND patients than in HC. The levels of aluminum and lead were significantly higher in non-ND patients than in HC. Obviously, the levels of gadolinium were significantly more elevated in ND and non-ND patients than in HC due to many MRI performed. No ND patient was affected by a Sn intoxication.

Interestingly, we found that non-ND patients displaying elevated levels of some toxic metals were affected by the following diseases: cardiovascular diseases, fibromyalgia, rheumatoid arthritis, and peripheral neuropathies.

We then compared the levels of toxic metals measured in urine samples following a chelation test in ND, non-ND, and ALS vs. HC. With an ANOVA test, we could appreciate that Al was significantly higher in ND, non-ND, and ALS, with respect to HC. Moreover, Pb was higher in ALS patients with respect to HC.

### 3.4. Reduction of Poisoning Following Chelation Therapy

Poisoned patients who underwent chelation therapies exhibited a significant reduction of toxic-metal levels (data not shown), as previously reported [16], accompanied by a consistent alleviation of related symptoms (headache, paresthesia, tingling, difficulty to walking, memory and visus loss, hypertension, and asthenia) [9]. In particular, ALS patients displayed improved weaknesses, as well as upper and lower motor dysfunctions. The results here described are superimposable to the previous ones [16]. ND patients displayed a reduction of toxic-metal levels following about 20–30 chelation therapies and constantly ameliorated with repeated chelations. As an exemplification, we report the case of a patient affected by MS. Figure 3 shows the toxic-metal levels following the chelation test. High levels of gadolinium, owing to MRI, cadmium, and lead, are evident. The patient underwent 60 chelation therapy applications over a 20-month period. Figure 4 shows a dramatic reduction in gadolinium levels (from 82 microg/g creatinine to 17 microg/g creatinine), as well as reduced lead and cadmium levels. Of note, reductions over time of gadolinium levels permits the elimination of those toxic metals present in minimal quantities unaffected by previous chelation therapy, such as mercury and cesium.

## 4. Discussion

The involvement of toxic substances in the pathogenesis and progression of ND is widely debated. The neuroinflammation hypothesis regarding the link between air pollution and ND supports the concept that inhaled pollutants activate the microglial production of cytokines and reactive oxygen species in the brain that can progressively damage the neurons [17]. The contributing role of miRNA alterations in the pathogenesis of neurodegenerative processes in response to environmental stimuli, such as metals and pesticides, has already been described [18]. Although genetic mutations are known to be responsible for the onset of ND, new evidence suggests that ALS, AD, and PD are caused by complex gene-environment interactions involving metal neurotoxicity [19]. Even excessive exposure to essential metals, such as iron and manganese, might lead to pathological conditions, such as neurodegeneration through impaired homeostasis, in essential metal metabolisms [20]. High levels of copper, manganese, and iron, responsible for Wilson’s disease, manganism, and hemochromatosis, respectively, exert an important role also in the pathogenesis of ND participating in the formation of α-synuclein aggregates in intracellular inclusions in the central nervous system (CNS); in particular, the accumulation of iron is responsible for PD and AD [21]. On the other hand, transition metals act as catalysts in oxidative reactions, causing oxidative tissue damage. In particular, redox-active metals, such as iron, copper, and chromium, undergo redox-cycling, whereas redox-inactive toxic metals, like lead, cadmium, and mercury, deplete major cell antioxidants, such as thiol-containing antioxidants and enzymes [22,23,24]. More recently, mercury and lead, in a concentration-dependent way, have been shown to induce an increase in amyloid beta protein (Aβ42) misfolding and aggregation with toxic properties, suggesting their implication in AD [25]. The potential relationship between mercury exposure and AD has also been further described [16]. In a neuronal cell human model, exposure to cadmium has highlighted gene deregulation, carcinogenicity, perturbations of essential metals, interference with calcium regulation, and other effects involved in neurodegeneration [26]. Moreover, metal-induced neurotoxicity has been linked to autophagic dysfunction, as the deficient elimination of abnormal or toxic protein aggregates can promote cellular stress, failure, and death [27]. Some heavy metals (e.g., lead, cadmium, and aluminum) used at subtoxic concentrations can lead to oligodendrocyte dysfunction, especially when oligodendrocytes are cocultured with neurons. The most important dysfunctions relate to imbalanced intracellular calcium ion regulation, altered lipid formation, and imbalanced myelin formation [28]. Aluminum toxicity in humans due to chronic inevitable exposure has already been described [29,30]. Furthermore, mercury neurotoxicity seems to be potentiated by the presence of apolipoprotein E4 [31].

We previously demonstrated that, unlike in HC, high levels of toxic metals are present in ND patients and in non-ND patients [16]. We extend this notion, showing that ND patients are affected by higher levels of each considered toxic metal compared with HC (Figure 2), except for thorium, which was found only in one HC. Moreover, here, we demonstrate that the profile of toxic metals is similar in ND and non-ND groups, which both display high levels of lead, cadmium, gadolinium, cesium, aluminum, and nickel, and that EDTA chelation therapy is effective in removing metal burdens. Of note, all patients (24/24) in the group of ALS were intoxicated by lead and cadmium; moreover, 14/24 were intoxicated by Al and 16/24 by Cs. Of note, Pb was significantly higher in ALS patients with respect to HC. Maximal levels of lead and cadmium intoxications were reached in non-ND patients, which we have examined in a greater number than before [16]. This observation is not surprising, because our cohort of non- ND patients was affected by either cardiovascular disease or by fibromyalgia, both diseases whose pathogenic mechanisms might be related to toxic-metal burdens [10,32,33] and whose detrimental effects we have shown to impact on and damage not only neurons but, also, other cell types—in particular, endothelial cells [34] Patient intoxication is a chronic event and requires several chelation therapy applications to reduce the toxic-metal burdens and improve symptoms (Figure 3 and Figure 4). The elevated levels of Gd in ND and, also, in non-ND patients are important to be considered as responsible for neurotoxicity. Our therapeutic approach relies in the administration of two grams once a week. After ten chelations, the therapy is able to reduce all toxic metals and to improve the patient symptoms, listed in the Results section; subsequent chelations progressively improve their reductions, often reaching physiologic levels. In particular, we measured in patients treated with EDTA chelation therapy the blood levels of Na, K, Mg, Cl, Ca, P, and Fe, which were not affected (data not shown). The therapy is well-tolerated and not associated with side effects.

The beneficial effect exerted by EDTA therapy can be supported by our experimental evidences, suggesting that EDTA may revert cellular endothelial damage induced in vitro by the cytokine TNF-alpha [35]. Moreover, in patients treated with EDTA chelation therapy, the levels of ROS in blood samples, as well as of oxLDL, were reduced and associated with an increase of the total antioxidant capacity, overall suggesting the role of EDTA as an antioxidant compound [36,37]. Finally, in patients affected by ND, the low levels of free glutathione (GSH) in erythrocytes were increased by EDTA chelation therapy, reaching those of control patients [38].

## Figures and Tables

**Figure 1 biomedicines-08-00269-f001:**
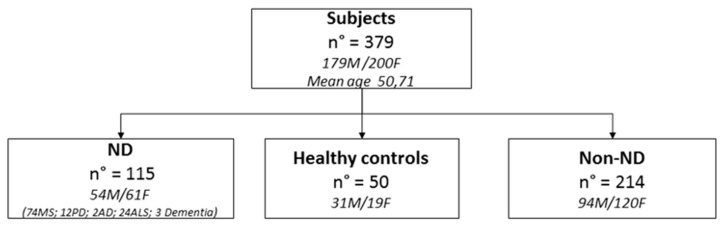
Characteristics of enrolled subjects. ND: neurogenerative diseases, MS: multiple sclerosis, AD: Alzheimer’s disease, and ALS: amyotrophic lateral sclerosis.

**Figure 2 biomedicines-08-00269-f002:**
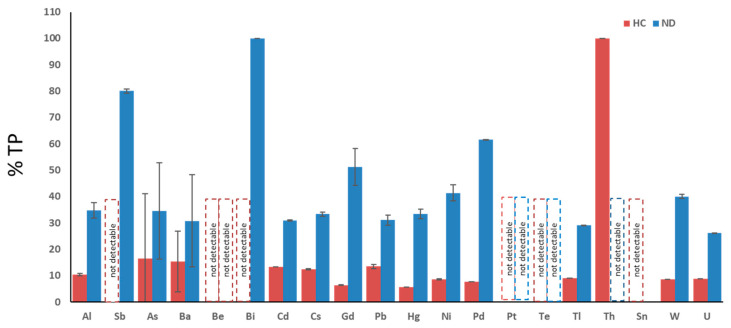
Percentage of total patients (TP) affected by toxic-metal burdens (mean ± SEM). ND = patients affected by neurodegenerative diseases. HC = healthy controls. Aluminum (Al), antimony (Sb), arsenicum (As), barium (Ba), beryllium (Be), bismuth (Bi), cadmium (Cd), cesium (Cs), gadolinium (Gd), lead (Pb), mercury (Hg), nickel (Ni), palladium (Pd), platinum (Pt), tellurium (Te), thallium (Tl), thorium (Th), tin (Sn), tungsten (W), and uranium (U).

**Figure 3 biomedicines-08-00269-f003:**
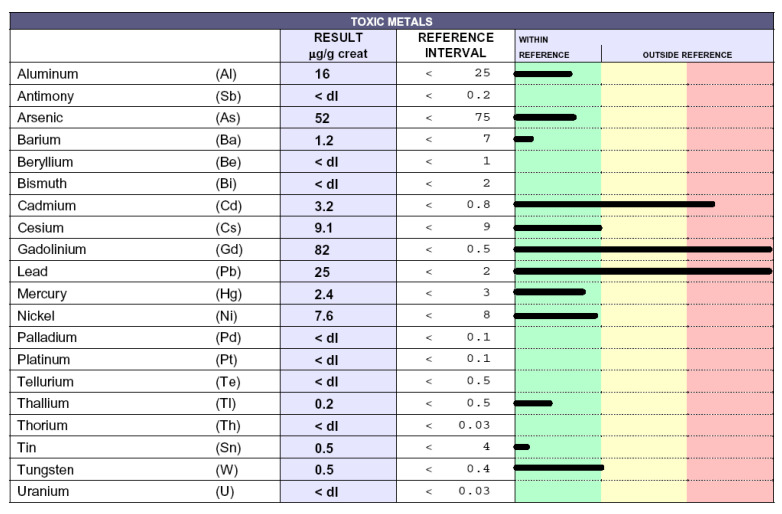
Toxic-metal levels measured by inductively coupled plasma mass spectrometry in patient urine collected during the 12 h following the ethylenediaminetetraacetic acid (EDTA) challenge (chelation test) reported in µg/g creatinine. The black lines indicate the levels of each toxic metal. Whitin the green column the values are considered normal, while in the yellow and in the red columns high and very high values are reported, respectively. The 42-year-old male patient was affected by multiple sclerosis and was a smoker.

**Figure 4 biomedicines-08-00269-f004:**
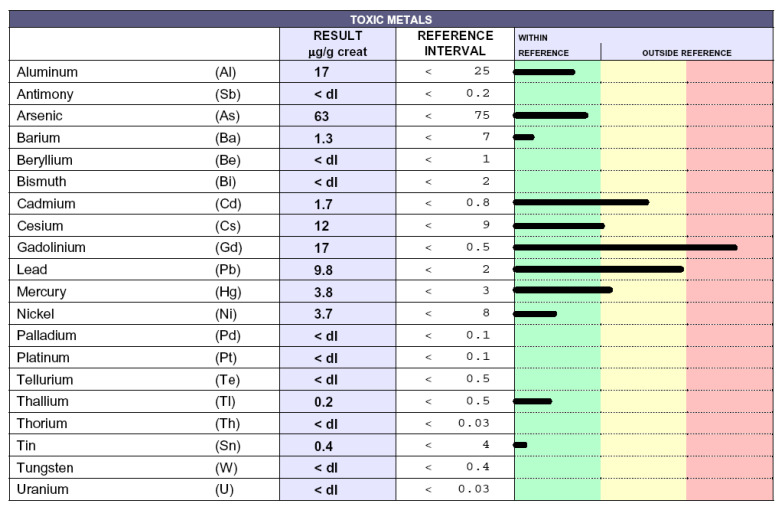
Toxic-metal levels measured by inductively coupled plasma mass spectrometry in the patient urine collected over 12 h reported in µg/g creatinine. The black lines indicate the levels of each toxic metal. Whitin the green column the values are considered normal, while in the yellow and in the red columns high and very high values are reported, respectively. The patient was affected by multiple sclerosis and underwent 60 chelation therapy applications over a 20-month period.

**Table 1 biomedicines-08-00269-t001:** Toxic-metal burden values assessed in the urine samples taken from patients following the chelation test. The cutoff represents the limit values of toxic metals, as higher values indicate toxicity. Patients with toxic-metal values above the cut-off are reported in the column 3 and indicated as A for each toxic metal in the total population (TP = all patients examined). The percentage of intoxication by each toxic metal with respect to the TP is reported in column 4. Columns 5 and 6 respectively show the mean and standard error of the mean (SEM) of the metal level values above the cut-off. Columns 7 and 8 show the number of ND patients burdened with each toxic metal and the percentage of those patients vs. A. Columns 9 and 10 show the mean values of toxic-metal levels above the cut-off and the SEM in ND patients. Columns 11 and 12 show the number of ALS patients affected by toxic-metal burdens (i.e., levels of toxic metals above the cut-off) and their percentage vs. A. Columns 13 and 14 show the mean values and SEM of toxic-metal levels above the cut-off in ALS patients. Columns 14 and 15 show the number of patients affected by each toxic-metal burden and their percentage vs. A in non-ND patients. Columns 16 and 17 show the mean values of toxic-metal levels above the cut-off and SEM in non-ND patients. Columns 18 and 19 show the number of patients affected by toxic-metal burdens relative to each toxic metal and their percentage vs. A in HC patients. Columns 20 and 21 show the mean values and SEM of toxic-metal levels above the cut-off in HC patients. * *p* < 0.05 HC vs. A and ° *p* < 0.05 HC vs. non-ND. A = number of patients with toxic metal levels > cut-off for each toxic metal. TP = total population, e.g., all patients examined. ND = patients affected by neurodegenerative diseases. Non-ND = patients affected by non-neurodegenerative diseases. HC = healthy controls. ALS = patients affected by amyotrophic lateral sclerosis.

	Cutoff	N° Pz > Cutoff (A)	% TP	Mean (>Cutoff)	SEM	N° Pz > Cutoff ND	% Vs A	Mean (>Cutoff)	SEM	N° Pz > Cutoff ALS	% Vs A	Mean (>Cutoff)	SEM	N° Pz > Cutoff non ND	% Vs A	Mean (>Cutoff)	SEM	N° Pz > Cutoff HC	% Vs A	Mean (>Cutoff)	SEM
Aluminum	25.00	135	36.4	41.93	1.74	47	35	46.70	2.90 *	14	10	56.86	2.21	65	48	47.67	2.11 °	14	10	32.43	0.53 *^,^°
Antimony	0.30	5	1.6	1.25	0.27	4	80	1.53	0.73	1	20	0.07	nd	1	20	1.80	nd	0	nd	nd	nd
Arsenic	108.00	55	15.3	252.07	13.85	19	35	269.47	18.34	1	2	77.95	nd	25	45	317.27	21.35	9	16	288.89	24.73
Barium	7.00	13	3.7	76.43	7.42	4	31	52.83	17.49	2	15	3.05	23.43	6	46	43.78	48.45	2	15	40.50	11.53
Beryllium	1.00	0	nd	nd	nd	nd	nd	nd	nd	0	0	0.00	nd	0	nd	nd	nd	0	nd	nd	nd
Bismuth	10.00	1	0.3	11.00	2.87	1	100	11.00	nd	1	100	0.00	nd	0	nd	1.20	nd	0	nd	nd	nd
Cadmium	0.80	346	92.9	3.49	0.16	107	31	3.96	0.21 *	24	7	3.08	0.16	179	52	4.30	0.14	46	13	2.52	0.08 *
Cesium	9.00	177	47.5	14.63	0.53	59	33	16.00	0.82 *	16	9	8.78	0.64	86	49	16.78	0.53	22	12	12.88	0.25 *
Gadolinium	0.30	172	45.9	31.19	5.41	88	51	41.55	6.96	7	4	58.83	0.81	65	38	8.48	4.34	11	6	2.62	0.18
Lead	2.00	370	99.7	26.76	1.56	115	31	28.00	1.91 *	24	6	22.92	1.75	192	52	38.79	1.53 °	50	14	20.16	0.77 *^,^°
Mercury	3.00	18	5.0	7.58	0.66	6	33	7.17	1.78	3	17	1.55	0.12	11	61	7.53	2.57	1	6	4.60	nd
Nickel	10.00	58	16.1	16.84	0.99	24	41	21.50	3.07 *	7	12	8.80	0.96	29	50	21.93	0.58	5	9	12.00	0.16 *
Palladium	0.30	13	3.4	0.48	0.04	8	62	0.48	0.05	1	8	nd	nd	5	38	0.65	0.03	1	8	0.40	nd
Platinum	1.00	2	0.5	10.65	9.35	nd	nd	nd	nd	0	0	nd	nd	2	nd	nd	9.35	0	nd	nd	nd
Tellurium	0.80	0	0.0	nd	nd	nd	nd	nd	nd	0	0	nd	nd	0	nd	nd	nd	0	nd	nd	nd
Thallium	0.50	55	14.8	1.20	0.06	16	29	1.33	0.12	8	15	nd	0.03	31	56	1.13	0.12	5	9	0.72	0.02
Thorium	0.03	1	0.3	0.08	0.04	nd	nd	nd	nd	0	0	nd	nd	0	nd	nd	nd	1	100	0.08	nd
Tin	9.00	4	1.1	24.00	1.47	nd	nd	nd	nd	0	0	nd	nd	3	75	48.00	2.29	0	nd	nd	nd
Tungsten	0.40	35	9.5	1.18	0.18	14	40	2.06	0.82	7	20	nd	1.14	19	54	2.88	0.04	3	9	0.60	0.03
Uranium	0.03	23	6.3	0.39	0.24	6	26	0.25	0.076	1	4	nd	nd	15	65	0.31	0.299	2	9	0.09	0.004

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
