# Peer review of "EDTA Chelation Therapy in the Treatment of Neurodegenerative Diseases: An Update"

_biomedicines, 2020, doi:10.3390/biomedicines8080269_

Round 1

Reviewer 1 Report

In this paper by Fukgenzi et al.m the authors describe the role played by toxic metals in neurodegenerative diseases. An enormous amount of data must have been obtained from these ICP-MS studies which that authors have attempted to present. This results in a poorly written paper where the important results are lost in the rambling text.

Introduction

Mitochondrial dysfunction and gut microbiota are discussed although the actual relevance of these 21 toxic metals in their function is not discussed. Do we know what biological function would be altered by an overload of each of these 21 toxic metals?

It is true that there is a link between inflammation (neuroinflammation) and metal function impairment although such studies have focused mostly on iron and copper.

The selection of the patients for the study are primarily ALS. It was a shame that the study design did not gain an equal number of ALS, PD and AD patients which should have been relatively easy to do. With the current data it would have been more appropriate to concentrate on ALS patients.

ALS initiates during mid adult life such age must be an important factor in toxic iron deposition. It is therefore necessary to plot both ND patients results and controls against age.

Results

I am unclear as to which results are presented. In the methodology it is stated that the patients underwent chelation therapy every 3 months on 10 occasions. Which occasion is presented in the results?

Also what does ‘spontaneously chose’ mean.?

Figure 2 is ridiculous as no standard deviation is presented..

Table 1 shows an abundance of data which is difficult to comprehend,. It would have been more appropriate to investigate only ALS patients.

How were the cut off values determined?

Line 9 and 10 do investigate only ALS but age will be an important determinant of urinary metal excretion

There is a long rambling   presentation of the results of Table 1. This needs to be cut dramatically such that the most significant results are given.

Did the authors investigate whether their subjects were smokers (would increase cadmium) and whether they were vegans etc.

One case of chelation in an MS patient are presented. What was the age of the patient? Could the increased lead and cadmium be attributable to smoking and his mode of work?

In the discussion the authors suggest that it maybe air pollution that is an important factor in the activation of microglia and neuronal damage. If this is the reason for these studies this should have been presented in the Introduction.

I am unsure why chromium is mentioned as an important factor in redox cycling. Is this true in a biological context?.

In UK iCP-MS has been used in routine laboratories for over 10 years and would not be referred to as a new method.

This paper needs to be totally rewritten with emphasis on ALS patients such that a clear message is given. Also what happened to each patient’s toxic metal load over the 10 chelation therapy regime?

Author Response

Thank for your careful comments.

Before beginning the point by point responses, we would like to anticipate some considerations.

The paper represents an extension and an update of previously obtained results on the usefulness of EDTA chelation therapy in the treatment of neurodegenerative and non-neurodegenerative diseases characterized by presence of toxic metal burden in the patients (see ref.12). Indeed, the role of toxic metals as cause of cell and particularly neuron damage as well as the beneficial effects of the therapy with EDTA in their removal have been extensively reported in the papers cited in the present work.

Introduction

Mitochondrial dysfunction and gut microbiota are discussed

Mitochondrial dysfunction and gut microbiota alterations are possible causes of neuron death. All 20 toxic metals induce metabolic alterations in cells and particularly in neurons and are responsible for the increase of free oxygen radical production, affecting biological function of multiple organs as reported in our previous paper (Ferrero, 2016). These concepts are now added in bold in the Introduction.

It is true that there is a link …..It is likely that iron and copper has a link with neuro-inflammation because Fe++ and Cu++ promote Fenton reaction only when their levels  are  higher than physiologic levels. To this notion, we and others add evidence that the presence of toxic metals has a link with neuro-inflammation always, independently of their levels.

The selection of the patients….The focus of our study are all the ND patients. We are conscious that the cohort of ALS patient could be of particular interest, given their complex, partially unknown, pathogenesis. In this update we were able to recruit an adequate number of ALS patients, comparable with that of MS patients studied in the previous paper (Fulgenzi et al., 2017)

ALS initiates during mild adult life…. In our experience and in our country the age of ALS patients is also very low and do not correlate with iron deposition. Moreover, iron deposition is not the object of our study because iron is not considered a toxic metal.

Results

I am unclear as to which results are presented…..The results relate to toxic metal burden studied following “chelation test”, as reported in the Results Section. Methodology (see Methods) describes both chelation test and chelation therapy which was performed by weekly treatment with EDTA intravenous infusion. Chelation test has been repeated after 10 subsequent chelation therapies.

Also what does ‘spontaneously chose’ mean? We agree that this sentence can be miss-understood. We now replace it with the new sentence “all patients gave their consent to the single treatment with “….

Figure 2 is ridiculous as no standard deviation is presented… Data are expressed in percentage; in the present version of the paper we add the standard deviation as you requires.

Table 1 shows an abundance of data which is difficult to comprehend…..Table 1 is very complex but we prefer make available all data related to all the considered patients’ groups, to allow the readers to compare them and to extrapolate data of interest.

How were the cut off values determined?

Reference ranges were evaluated based on literature, patient’ data obtained by Doctor’s data and other laboratories, For toxic metals, in absence of lower limit of acceptability for toxic elements, the stated levels on the report reflects what is found in 95% of the population and is plotted in the green area. Values plotted in the yellow area represent that which may be seen in 99% of the population. Values plotted in the red indicate that the levels exceeds that found in 99% of the population and may be clinically significant.

Line 9 and 10 do investigate only ALS but age ….To avoid this bias, we have normalized metal values in urinary excretion with those of the creatinine levels

There is a long rambling presentation of the results of Table 1….. We agree that the large amount of data can be confusing. Our general aim, however, is to share our data with clinicians and researchers expert in the field to give and receive suggestions.

Did the authors investigate whether their subjects were smokers…. We agree that lead and cadmium are significantly higher in smokers and that these correlations could be of great interest. However, our major aim was to reduce metal intoxication regardless of their causative agents, including life style.

One case of chelation in an MS patient are presented …The age of MS patient is 42, and he/she was a smoker, as now reported in the legend of Figure 3. The patient was selected as a representative one to confirm the efficacy of the chelation therapy.

Discussion

In the discussion the authors suggest that it maybe air pollution that is an important factor ….In the present version of the introduction we  report air pollution among the direct causes of  neuron damage.

I am unsure why chromium is mentioned as an important factor in redox cycling… We now report two recent references supporting that Chromium is involved in redox cycling (ref n 18 and 19)

In UK iCP-MS has been used in routine laboratories for over 10 years….. New has been deleted in the text.

This paper needs to be totally rewritten with emphasis on ALS patients such that a clear message is given…..In this paper we  incorporate all ND patients to provide a consistent evidence that toxic metals are involved in their pathogenesis and that  EDTA chelating agent is efficient in removing detrimental metals.  Chelation therapy reduces toxic metal burden after 10 chelations; subsequent chelations progressively improve their reduction, as in Discussion.

Reviewer 2 Report

Overall, the authors investigated the impact of EDTA on metal chelation in ND. While this study provides an interesting observation, it has substantial issues that prevent from publication. Comments are provided below.

Issues:

  1. Introduction
    1. Intro lacks the information about molecular link between metals and ND.
    2. Background knowledge of EDTA could help.
  2. Methods
    1. What is the rationale for selecting 21 metals?
    2. There are at least three groups to compare, and therefore ANOVA would be an appropriate stat. Also, do the data follow normal distribution? If not, a nonparametric test should be used.
  3. Results
    1. Figure 2: Th level is higher in HC than in ND. Is this because of accidental exposure or erroneous measurement? Please discuss. It could be an unknown reason and could happen only in one individual as an outlier, but then there is the same case (one patient data) in Figures 3 and 4, posing an issue of replication.
    2. Lines 79-93: These sentences can be placed into the Table legend.
    3. Table 1: There is no considerable difference in metal levels between ND and non-ND. This does not match what the authors concluded.
    4. Line 75-193: This section is hard to follow and too many metals are described in one paragraph. I would recommend the authors use sub-heading and provide results for each metal.
    5. Figures 3 and 4: This is from one patient data and therefore does not provide meaningful results.
    6. Are endogenous metals (Fe, Zn, Cu, Mn) levels different between ND and HC and affected by EDTA treatment?
  4. Discussion
    1. The first paragraph in Discussion seems like a part of Intro. See comment on Intro above.
    2. It is unclear why non-ND individuals show significantly higher metal levels than HC. Some discussions are provided, but they do not appear to be related to brain or neuroinflammation. It would be more convincing to see statistical differences between ND and non-ND if the research focus was brain metal chelation in ND.
    3. Urine metal excretion does not necessarily indicate brain metal status. The conclusion seems a stretch.

Author Response

Thanks for your careful revision and suggestions

Introduction

1) Molecular link between metals and ND has been now reported in bold in the Introduction, 2) a background knowledge on EDTA has now been added in both Introduction (58-60) and Discussion (lines 74-81).

Methods

1) The 20 metals were selected because they are the widely recognized as toxic metals

2) ANOVA has not been used because we compared only two group (HC vs ND figure 2) as reported in the Methods Section.

Results

1) In our results toxic metals are constantly higher in ND than in HC patients. We consider the Thorium as an exception, possibly due to accidental exposure as now described in Results (line 78).

2) Lines 87-105 are now inserted in the table1 legend

3) Non-ND patients were affected by important diseases, as cardiovascular diseases, fibromyalgia, etc. which are associated with accumulation of toxic metals, as now reported in bold in the Results section.

4) Sub-heading for each toxic metal was now inserted in the text.

5) We agree with you. The case reported in Figure 3 and 4 is a representative case and we aim to analyze and add other cases hereafter. 

6) A very good question. We measured in patients treated with EDTA chelation therapy the blood levels of Na, K, Mg, Cl, Ca, P and Fe which were always normal (data not shown), as now added in the discussion. However, Fe, Zn, Cu and Mn are not considered toxic metals and we have not evaluated their differences between ND and HC patients.

Discussion

1) We have accordingly modified both Introduction and Discussion-

2) It is unclear why non-ND individuals…In Results point 3 we partially respond to this question.

It would be more convincing to see statistical differences between ND and non-ND if the research focus was brain metal chelation in ND… Your comment is a good point, however it goes beyond our aims and we have not available results on brain status.

3) We agree, urine metal excretion is indicative of the total toxic metal burden of the patients with no particular reference to brain.

Reviewer 3 Report

The manuscript describes the effect of EDTA chelation therapy in the treatment of ND. In general, the presentation of the manuscript must be improved. Below there are some specific comments and suggestions that should be attended before accepting this MS for publication in this journal.

  1. Add p-value to each comparison of metal level between HC and ND in figure 2.
  2. Please give details of specific ND with the metal level before and after EDTA chelation therapy.
  3. There are some side effects of EDTA. Please discuss these.
  4. The discussion is superficial. Discuss the advantages of your findings.
  5. The resolution is poor for Figures 3 and 4 and Table 2.

Author Response

Thanks for your careful revision and suggestions

1) p-value is reported.

2) We provide evidence of decreased levels of 20 toxic metals before and after the chelation therapy only in a ND patient as a representative one. In all ND patients the alleviation of clinical symptoms after the therapy is well monitored as reported in Results section, page 8, lines 2-8.

3) No side effects were observed by EDTA treatment in our conditions (use of 2g once a week) as now discussed.

4) Our findings are now discussed.

4) Resolution of figure 3 and 4 and table 2 has been improved

Round 2

Reviewer 1 Report

The authors have addressed many of my comments.

However I am still unclear as to which specimen was analysed after EDTA therapy. was it the first, the 10th or what?

Also was there any improvement in clinical symptoms in the ALS patients after EDTA therapy.

Their comment to me 'that iron is not considered to be a toxic metal' shows a complete ignorance of neurodegenerative diseases where the excess iron is responsible for much of the damage!!

The paper may be of interest to researchers involved in chelation therapy, but the potential use of EDTA in the treatment of neurodegenerative disease is a non-starter because of its lack of specificity. 

Author Response

The results reported in Table 1 are related to toxic metal levels measured in urine samples following the chelation test, e.g. after the first EDTA therapy (now added in red in the results section as comment to table 1)

EDTA chelation therapy was able to improve ALS  patient’s weakness as well as upper and lower motor dysfunction. The concept has been added in red in the results

We are conscious that high levels of iron can cause neurodegenerative diseases, but iron is not considered a toxic metal at physiologic levels. A new cited paper (in red) supports this concept and has been cited in the Discussion section.

This is a controversial issue. From our experience we have learned that in ND, characterized by a complex pathogenesis, EDTA chelation therapy has the advantage to remove some causes (toxic metals) of neuron damage [Ref 1] and in turn can specifically alleviate related symptoms.

Reviewer 2 Report

The manuscript has been improved, but there are still significant deficiencies. First, the authors have not responded to the reviewer’s comments point-by-point, and several comments by the reviewer were ignored. Second, the authors have not adequately addressed the reviewer's comments. For example, ANOVA should be performed based on the experimental design, and the simple t-test would provide incorrect results, often false-positive statistical significance. The comment on normal distribution was ignored. The authors are encouraged to consult statisticians to justify the use of the t-test. Background information about EDTA is not adequate. Increased metal levels in non-ND patients are not well discussed, and the results appear to indicate that these non-ND disease conditions significantly increase the toxic metal levels, which are somewhat against the authors' conclusions. The authors are encouraged to clarify this finding and expand the discussion to support their conclusions. An increasing body of evidence suggests that elevated levels of transition metals (Fe, Cu, Mn) are toxic and closely associated with ND (Copper overload in Wilson’s disease, manganism, etc), and it is relevant to discuss this point in the paper as it relates to toxic metals. Also, it is unclear how anticipated EDTA's action (ROS in blood, antioxidant, etc) could benefit ND patients. More discussion with respect to ND would help the readers. 

Author Response

First: the authors have not responded to the reviewer’s comments point-by-point, and several comments by the reviewer were ignored.

We now provide more exhaustive answers. In particular:

The comment on normal distribution was ignored:  Our data follow a NORMAL DISTRIBUTION

ANOVA should be performed based….

As the reviewer suggests, we have consulted an expert statistician. Results in Table 1 compare values of toxic metal burden obtained after chelation test with values herein indicated as A (=total intoxicated population with presence of toxic metals>cut-off) within each group of patients. ANOVA test is not adequate to compare only two groups.

To meet your requirement, we have performed ANOVA test, comparing the 3 groups, ND, non-ND and ALS with HC. ANOVA test  highlights important findings: aluminium was significantly higher in ND, non-ND and ALS patients than in HC. Moreover lead was significantly higher in ALS patients than in HC. These findings are added in the text, above the Table 1 (and could be at your disposal, in a confidential form, if required).

Background information about EDTA is not adequate: to meet this comment, we now add some information in the introduction, in red.

Increased metal levels in non-ND patients are not well discussed… We now add a comment in discussion in red.

 An increasing body of evidence suggests that elevated levels of transition metals (Fe, Cu, Mn)…Transition metals are toxic only when present at high levels ( a new paper has been reported in the Discussion, Mezzaroba L et al reference number 21) but we have studied only the metals which are toxic  yet at low levels

Also, it is unclear how anticipated EDTA's action…. We have previously demonstrated that EDTA chelation therapy exerts antioxidant activity and reduces ROS levels (Fulgenzi et al, Biomed Res Int 2014, now cited in the discussion)

Round 3

Reviewer 2 Report

Reviewer's issues have been addressed.